# A Review of Microalgae- and Cyanobacteria-Based Biodegradation of Organic Pollutants

**DOI:** 10.3390/molecules27031141

**Published:** 2022-02-08

**Authors:** Hussein El-Sayed Touliabah, Mostafa M. El-Sheekh, Mona M. Ismail, Hala El-Kassas

**Affiliations:** 1Biological Sciences Department, Faculty of Science, King Abdulaziz University, Jeddah 21589, Saudi Arabia; 2Botany Department, Faculty of Science, Tanta University, Tanta P.O. Box 31527, Egypt; mostafaelsheikh@science.tanta.edu.eg; 3National Institute of Oceanography and Fisheries (NIOF), Alexandria P.O. Box 21556, Egypt; mm.esmail@niof.sci.eg (M.M.I.); halayassin12@yahoo.com (H.E.-K.)

**Keywords:** algae, bioremediation, phycoremediation, azo dyes, herbicides, pesticides

## Abstract

This review proposes a new bioremediation method based on the diverse functionalities of algae. A greenway for cleansing wastewater is more ecologically friendly and environmentally sustainable than prior methods with other bacteria. New bioremediation technology employing algae and cyanobacteria for the removal of a wide range of organic contaminants is reasonable and has great potential. The prevalence of organic contaminants in aquatic habitats may endanger the health and well-being of several marine creatures. Agriculture, industry, and household trash are just a few of the human-caused sources of organic pollutants that contaminate waterways around the world. Before wastewater can be released into waterways, it must be cleaned. Algae-based wastewater treatment systems are becoming increasingly popular because of their environmental sustainability and lack of secondary pollutants. According to the kind of pollutant, the physicochemical properties of wastewater, and the algal species, algae and cyanobacteria can absorb and accumulate a wide spectrum of organic pollutants at different rates. In addition, phytoremediation is a cost-effective alternative to conventional treatments for degrading organic contaminants. Phycoremediationally produced algal biomass may also be an important part of the bioenergy value chain. This article focuses on microalgae and cyanobacteria species, which may remove many organic contaminants from water systems.

## 1. Introduction

Water pollution has emerged as one of the most serious environmental problems on a global scale due to the rapid growth of urbanization and industrialization [1]. Pollution occurs in water systems when a large number of organic substances from domestic and agricultural sewage (raw or treated), urban runoff, and industrial waste combine and have a detrimental effect on water quality due to the toxicity, bioaccumulation tendency, and persistence of such substances, as well as their susceptibility to undergo long-range atmospheric transport and deposit [2]. Additionally, these pollutants may be immobilized and bioaccumulated in sediments or transformed and activated in aquatic systems [3].

Microalgae and cyanobacteria represent a possible new option for the bioremediation of distillery wastewaters since they have trophic independence for nitrogen and carbon. However, because they are light-dependent reactions, dilution of the colored effluents to be treated is required to avoid light blockage [4]. In this review, for a number of aromatic contaminants and structurally related chemicals, we highlight recent successes in distillery wastewater treatment research. We also consider the ability to metabolize phenolics, such as lignin or tannins, as well as the ability to break down melanoidins, the major colored chemicals in slops. We also document certain enzymatic aspects of phenol and melanoidin biodegradation. Through large-scale microalgal production, different routes for improving wastewater treatment technology have been reported. Along with genetic-engineering research, the technoeconomic feasibility and main commercial-production obstacles have been overcome. A biorefinery strategy combining integrated biology, ecology, and engineering would result in a microalgal-based technology that may be used in a variety of applications [5].

## 2. Organic Pollutants

### 2.1. Organic Hydrocarbons

The primary groups of organic hydrocarbon pollutants are polycyclic aromatic hydrocarbons (PAHs), polychlorinated biphenyls (PCBs), di-(2-Ethylhexyl) phthalate (DEHP), surfactants, and polychlorinated dibenzo-p-dioxins and dibenzofurans (PCDD/PCDF). They are hydrophobic chemicals that survive in water systems for a long time and are associated with sediments. Petroleum products are the primary source of hydrocarbon pollution in the aquatic environment [6]. These compounds typically reach marine environments via rivers, coastal waters, industry, and the atmosphere due to natural processes occurring during the earth’s biogeochemical cycles [7]. Additionally, these compounds are gently decomposable and resistant to hydrolysis in aerobic circumstances in bodies of water, as deprivation rates fall dramatically as the number of aromatic rings and molecular weight increase [8]. For instance, PAHs are classified as carcinogens; they are widespread environmental hazards due to their toxicity and slow breakdown in the environment [9]. Personal care products (PCPs) have various detrimental effects on aquatic organisms, including endocrine, developmental, and epigenetic systems, and directly impact human health [8]. Phenol is a major carcinogenic organic pollutant found in the effluent of many businesses, including petroleum refineries, with concentrations ranging from 13 to 88 ppm. It is hazardous to marine organisms at concentrations of 5–25 ppm. Traditional approaches are ineffective at removing phenol chemicals from water systems. However, biological methods appear promising due to the high removal effectiveness of the organisms used [10,11,12].

### 2.2. Organic Dyes

Chemical and electrochemical methods, such as adsorption and ultrafiltration, cannot remove organic dyes from water because they are not readily decomposable. Therefore, organic dyes are a major pollutant in water [13,14]. These pollutants also hurt aquatic photosynthesis, which is the basis of the food chain [15]. Anthraquinone, azo, and phthalocyanine are the three main categories of dyes, based on the chemical structures of their chromophore groups [16]. Cationic basic dyes are divided into two categories: dyes that are not dispersed in an ionic solution; and dyes that are anionic-direct, acidic, or reactive in nature [15]. The adsorption method for the decolonization process is an easy operation with low cost [17], whereas the adsorption technique is the most widely utilized method for decolonization due to its low cost and convenience of use [17].

Due to their wide spectrum of colors, azo (3000 different types) and anthraquinone dyes are the most extensively used dyes worldwide [18]. They can be found in many manufacturing industries, including textiles, plastics, and medicine. A large amount of high-color-content effluents, which show toxicity at relatively low levels, is blamed for harmful effects. These dyes are also extensively accumulated, resulting in eutrophication and limited reoxygenation capacity. The generation of poisonous amines during the decomposition of azo dyes is one of the most serious concerns. Therefore, most aquatic organisms die due to their toxicity [17].

### 2.3. Organometallic Compounds

The industrial use of organometallic compounds means that a large number of these compounds end up in waterways. The toxicity of organometallic compounds is little understood, despite being the most deadly substances on the market [19,20]. Occupational exposure to organotin compounds (OTCs) can result in cancer and other health problems. Despite this, they are widely utilized as antifouling agents in aquatic systems, increasing pollution and posing a severe concern for diverse ecosystems, even at extremely low concentrations [21]. Polyurethane foam and silicone manufacturing processes use organometallic compounds as catalysts [22]. Toxic to aquatic species, tributyltin is the most widely dispersed OTC toxin available today [21].

### 2.4. Pesticides and Herbicides

Pesticides have become a critical problem in many developed countries, causing surface and groundwater pollution. Fungicides (fungal destroyers), herbicides (weed killers), nematicides (parasitic nematodes), insecticides (insect killers), and rodenticides (vertebrate poisons), are the most widely used pesticides in agriculture for controlling a wide variety of insectivorous and herbaceous pests [23]. Besides being harmful, pesticides are commonly distributed in the environment because they are mobile, bioaccumulative, and persistent [24]. Their presence in water is harmful for ecosystems, the quality of drinking water, and human health [25].

## 3. Algae as an Organic Biodegradation

Algae play an important role in the biological treatment of wastewater, called “phycoremediation”. Algae can accumulate organic and inorganic toxic substances, as well as radioactive materials, in their cells. Consequently, they play important role in the self-purification of municipal, industrial, and agroindustrial wastewater. Moreover, they are the main producer in water systems, and many algae species flourish in water polluted with various types of organic waste by developing several detoxifying mechanisms, including biosorption, bioaccumulation, biotransformation, biomineralization, and in situ and/or ex situ biodegradation, as shown in Figure 1 [26]. Biosorption is a physicochemical process that is rather dependent on different mechanisms, including absorption, adsorption, surface complexation, ion exchange, and precipitation. Biosorption is an ideal process for the removal of various pollutants, e.g., textile dyes, phenolic compounds, and pesticides. In addition, it is an eco-friendly and cheaper method for removing contaminants [27]. The algal biodegradation process can occur either extracellularly; intracellularly; or a combination of both, where the initial degradation occurs extracellularly, and breakdown products are subsequently degraded intracellularly [28]. Biosorption is a series of activities that occur in the cell wall before anaerobic or aerobic biodegradation. It has good selectivity and efficiency (high performance and low cost). Biosorbents can be natural materials, such as marine algae or weeds, or industrial waste, such as activated sludge or fermentation waste [29], and represent the best method for the mechanical elimination of biological sludge. The dangers of biomass sludge depend on the materials absorbed or adsorbed. Using biosorption to remove organic contaminants, such as pesticides and phenols, was discussed in a review study by Aksu [15]. Removing dangerous and toxic pollutants from the environment, even if expensive, is a more ecofriendly alternative than leaving them.

We consider biosorption ecofriendly since it can not only adsorb but also degrade pollutants, such as azo dyes, into simple compounds. Algae and cyanobacteria can degrade pollutants through excretion of specific enzymes, such as azo-dye reductase, and convert dyes to simple, nontoxic compounds, such as NH_2_ and CO_2_. Little research has been conducted on the efficiency of phycoremediation in reducing nutrient levels in eutrophic lakes. On the other hand, Razak and Sharip [30] discussed the promise of algal-based approaches for the resolution of lake eutrophication. They reported that wastewater is the primary focus of most phycoremediation studies. Using microalgae to reduce fertilizer, pollution, and heavy-metal levels in lakes is a surefire way to improve water quality. Lake phycoremediation has received little attention from researchers. Therefore, the ecological and physiological features of algae should be studied further, especially as they relate to eutrophication. The effectiveness of phycoremediation must be studied in natural settings, such as lakes and ponds. Bioremediation uses organisms such as bacteria, fungi, and algae that can metabolically break down toxins. Nutrient enrichment promotes the growth of naturally occurring algae in all water bodies. These indigenous bacteria work similarly to those utilized in bioremediation processes when algae release oxygen into the air. Photosynthetic algae provide clean oxygen to the atmosphere, improving water quality, marine life, and biodiversity [31,32]. Phycoremediation preserves the natural food chain of lakes, making it the most efficient nutrient removal method. Algae and natural microbes work together to clean water. Algae use CO_2_ and nutrients to generate O_2_ [33,34], and no waste is created under either mixotrophic or heterotrophic conditions [35].

Phycoremediation is a relatively safe technology, since algae convert sunlight to useful biomass by utilizing nutrients such as phosphorus and nitrogen, the primary contributors to eutrophication [36]. Both living and nonviable algae were shown to be efficient at accelerating biodegradation and converting organic pollutants to simple molecules [37]. Many studies have considered the use of microalgae and cyanobacteria species for phytoremediation of organic pollutants in aquatic environments [38,39].

Many cyanobacterial species are distinguished by their strong ability to remove organic pollutants from aquatic environments as a result of their high photosynthetic activity, which results in the formation of huge biomasses. These biomasses contain a variety of bioactive chemicals that can be used in a variety of cost-effective applications [38,39].

Cyanobacteria and eukaryotic microalgae are distinguished based on their rapid growth in sterile conditions. Due to its high polyunsaturated-fatty-acid content, algal biomass was previously reported to be used for several purposes, such as biodiesel, biogas, and bioethanol [40]. The rate of removal and biodegradation of pollutants depends on their concentration, molecular weight, algal biomass, species, metabolic activity of algae, growth phase, and environmental conditions [14]. Numerous algae of various classes thrive and grow best in contaminated environments, including *Euglena viridis* (Euglenophyta), *Nitzschia palea* (Bacillariophyta), *Oscillatoria limosa*, *O. tenuis*, *O. princeps*, and *Phormidium uncinatum* (Cyanophyta) [41].

Algae have three specific ways to remove organic pollutants, including PAHs, from ecosystems. The first is by adsorption of PAHs on the algal cell wall of active groups by electrostatic attraction and complexation. The second is by bioaccumulation of pollutants inside the cells. Regarding this process, the biosorption and accumulation ability of algae is related to their functional groups, including the hydroxyl, carboxylate, sulfate, phosphate, and amino groups [42].

Meanwhile, the biosorption process involves a variety of chemical reactions occurring on the cell surface. These reactions include ion exchange with active groups on the algal surface, surface complexation reactions, chelation, and microprecipitation [43]. The third way to remove organic pollutants is to convert them into beneficial chemicals, such as carbon dioxide and water, using redox enzymes involved in enzymatic oxidation-reduction reactions [44]. Microalgae include many enzymes that contribute to the breakdown of various chemical compounds that cause cellular stress in microalgae [45]. These are complicated enzymes that span both phase I and phase II. Monooxygenases, dioxygenases, hydroxylases, carboxylases, and decarboxylases play an important role in the biodegradation of phase I enzymes [46] and are responsible for increasing the hydrophilicity of pollutants by adding or unmasking a hydroxyl group via oxidation, reduction, or hydrolysis reactions. Phase II contains an enzyme that catalyzes the conjugation of glutathione and glutathione-S-transferases with a diverse set of chemicals with electrophilic sites, resulting in the opening of the epoxide ring and protection against oxidative stress in the cell [47]. Laccase glycoproteins are extracellular enzymes and play a critical role in the microalgal biodegradation of different organic pollutants and either deplete reactions after that or undergo cross linking [48]. The other theory explains the algal biodegradation mechanism in an association with which traditional water treatment methods are expected to be replaced and considered to be a highly promising technique for the removal of different pollutants [49]. This includes two processes: (1) Biosorption, which takes place when pollutants in an aqueous solution are either adsorbed to form algal cell-wall components or onto organic substances excreted by algal cells, especially protein and extracellular polysaccharides that are secreted into the surrounding environment. This is a passive technique dependent on many factors like, such as the types of biosorbents, environmental conditions, and algal metabolic processes [50]. (2) Bioaccumulation through the removal of organic pollutants, depending on algal metabolic activity, also describes how pollutants enter a food chain via algae [51]. Among the harmful effects of pollution, eutrophication is the most common. High amounts of organic materials and decomposing organisms deplete oxygen in water and kill other species. The nutrients (NH_4_^+^, NO_3_^−^, PO_4_^3−^) in secondary effluents are the principal source of eutrophication in natural waters. These nutrients degrade water quality and harm aquatic ecosystems [52,53]. Therefore, before it is released into waterbodies, wastewater must be appropriately treated. Activated sludge (for nitrogen and phosphorous removal), electro flocculation, membrane filtration, electrokinetic coagulation, electrochemical destruction, ion exchange, CatOx treatment (catalytic oxidation), and disinfection (by ozonation, chlorination, or ultraviolet light) are some of the unit processes used to remove nutrients from wastewater [53].

Algae are characterized by the secretion of various enzymes involved in the biodegradation of several organic pollutants into less hazardous metabolites/moieties. They also possess catabolic genes for the degradation different pollutants [54]. Biosurfactant-producing algae may significantly enhance the biodegradation of hydrocarbon-polluted positions [51]. These biosurfactants are complex biomolecules, including glycolipids, lipoproteins, lipopeptides, neutral lipids, phospholipids, and fatty acids. These biosurfactants are amphiphiles, thereby increasing the solubility of hydrophobic pollutants in water to accelerate hydrocarbon bioremediation [55,56]. PCB biodegradation by algae is dependent on the physicochemical and physiological concentration and hydrophobicity of PCBs, the biomass and physiological capacities of the algae, as well as its membrane permeability [57].

### 3.1. Bioremediation of Organic Pollutants by Microalgae

Phycoremediation with microalgae species is an effective way to ensure environmental sustainability, as microalgae are good degraders of a variety of environmental toxins due to their high surface-area-to-volume ratio, quick metabolism, cost-effectiveness, and abundant availability [58]. Microalgae absorb and collect a variety of toxins through biological amplification, which can be passed up the food chain [59]. Algal bioaccumulation capacity is conditional on growing circumstances and depends on lipid content [60]. Numerous genera, including *Botryococcus*, *Chlamydomonas*, *Chlorella*, *Scenedesmus*, *Nitzschia*, and *Micractinium*, are recommended for wastewater treatment (Table 1). Additionally, these species have a high oil yield, which can be used to manufacture biodiesel [61,62].

#### 3.1.1. Dyes

Microalgae can degrade various types of dyes into carbon and nitrogen sources and then remove them from water, which leads to a reduction in eutrophication in the ecosystem. Numerous studies have been conducted, and researchers have documented the biodegradation of various dyes by a variety of microalgae species, such as *Chlorella* spp., *Scenedesmus* spp., and *Aphanocapsa* spp., depending on the dye’s molecular geometry, the algae species, and metabolism azo-reductase enzymes [65,70]. Microalgae was found to removes dyes in a variety of ways, including biosorption, bioconversion, and biodegradation. Laccase glycoproteins were found to be involved in a substantial amount of industrial colors and phenols [48]. These dyes are degraded by algal cells searching for nitrogen sources and then used in their growth, reducing eutrophication in the aquatic ecosystem [71]. Numerous *Chlorella* species were found to be capable of degrading azo dyes into simple organic compounds or CO_2_ via the metabolization of aromatic amines in combination with the breakdown of the azo link [72]. The activity of *Chlorella vulgaris* was detected to be associated with the degradation of more than 90% of azo dyes [64]. Additionally, the biodegradation activity of *Chlorella pyrenoidosa* and *Chlorella vulgaris* was confirmed concerning the degradation of more than 30 azo compounds into simpler aromatic amines or CO_2_, as shown in Figure 1. [73]. The ability of *Scenedesmus quadricauda* to degrade Reactive Blue 19 and Remazol Brilliant Blue R dye (RBBR) in a variety of aquatic environments was discussed in [74]. Furthermore, *C. ellipsoidea*, *C. kessleri*, *C. vulgaris*, *Sc. bijuga*, *Sc. bijugatus*, and *Sc. obliquus* were able to decompose both mono- and di-azo tartrazine [60,75]. Microalgae and cyanobacteria are used in phytoremediation to remove nutrients and toxins from wastewater and carbon dioxide from the air [53,76]. Photoautotrophic bacteria are desired since their use is ecofriendly and does not pollute the environment [77]. Many researchers use cyanobacteria in phycoremediation because of their ability to develop quickly and grow on non-arable land, as well as their low water and land requirements and photoautotrophic microorganisms [36,78,79,80]. Blue-green algae can consume CO_2_ and convert it to algal biomass, increasing levels of O_2_. These process can be used to producing biogas, biofuels, and in many other useful applications [80]. Furthermore, algae and cyanobacteria are capable of degrading pollutants by excretion of some enzymes which degrade the toxic compounds into simple nontoxic compounds. The ability of *Scenedesmus quadricauda* to degrade Reactive Blue 19 and Remazol Brilliant Blue R dye (RBBR) in a variety of aquatic environments was discussed in [74]. It is well known that algae are photosynthetic and can occupy any environment. Their crucial roles in the nutrient cycle and oxygen production are considerable in many ecosystems. Biomass from microalgae has been utilized as an adsorbent. Scenedesmus quadricauda is a newcomer to biosorption research. They use light instead of carbon sources like bacteria and fungus, allowing them to survive in the absence of organic carbon. Therefore, using metabolically active microalgal systems may be easier [74]. Recently, immobilized algae were used to solve this issue. Furthermore, *C. ellipsoidea*, *C. kessleri*, *C. vulgaris*, *Sc. bijuga*, *Sc. Bijugatus*, and *Sc. obliquus* were found to be capable of decomposing both mono- and di-azo tartrazine [65].

#### 3.1.2. Organic Hydrocarbon

Phytoplankton species are critical for maintaining a healthy water column by regulating the concentration of organic contaminants. They are capable of absorbing a variety of contaminants and accumulating a number of chlorinated hydrocarbons [69]. The bioremediation rate of phenanthrene (PHE) and fluoranthene (FLA) by *Nitzschia* sp. was found to be faster than that of *Skeletonema costatum* [81]. *Nitzschia* sp. bioremediated phenanthrene (PHE) and fluoranthene (FLA) more rapidly than *Skeletonema costatum* [81]. Some microalgae associated with crude oil may have remediation activity due to their ability to break down hydrocarbons into less complex chemicals without causing harm [82]. Benzo[a]pyrene (BaP) is a carcinogenic polycyclic aromatic hydrocarbon (PAH) with a high molecular weight; it can be degraded by many microalgae species, including *C. vulgaris*, *Sc. quadricauda*, *Sc. platydiscus*, and *Selenastrum capricornutum* depending on the algae’s density, cell-wall composition, and involved oxidation-reduction enzymes [61]. *Selenastrum capricornutum* has previously been shown to oxidize benzo[a]pyrene to sulfate ester and glucoside conjugates using a dioxygenase mechanism [83,84,85,86]. El-Sheekh [82] reported that *N. punctiforme* and *S. platensis* can grow under heterotrophic conditions using crude oil as a source of carbon (heterotrophic condition). Subashchandrabose [61] confirmed that microalgae *Chlorella* sp. are capable of degrading pyrene through the production of dihydrolipoamide acetyltransferase, leading to remediation of soils contaminated with pyrene. Pyrene biodegradation occurs at a species-specific rate that is mostly dependent on the concentration of the algal biomass [85,86]. The authors reported that pyrene’s growth-inhibiting effects on algae may be mediated through suppression of protein synthesis. Conversely, bacterial pyrene metabolites may enhance algal growth by stimulating DNA replication and protein synthesis. This research shows how algae and bacteria work together to degrade organic pollutants and reduce their toxicities to algae. Both *C. vulgaris* and *Sc. obliquus* have been discovered as potential biosystems for crude-oil degradation due to their rapid growth rates and capacity to biodegrade oil under heterotrophic conditions using waste oil as the sole carbon source [87,88,89]. El-Sheekh et al. [88] focused the ability of *S. obliquus* and *C. vulgaris* to grow under heterotrophic conditions using crude oil as sole carbon source. They proved the ability of both algae to degrade n-alkane and PAHs. This considered an innovative technique and an economically feasible method which could be used on a large scale. Ghasemi reported that algae can promote the degradation of pollutants either by improving the degradation capability of the microbial community or altering the pollutant directly. As a result of its tolerance to organic pollutants, *Chlorella* has colonized wastewater treatment systems. In addition to product selectivity, algae feature a recycling mechanism. Changing enzyme function by genetic engineering may produce useful enzymes. Previously, the green alga *Prototheca zopfii* was found to be capable of degrading around 49 ± 11% of saturated aliphatic hydrocarbons and 26.5 ± 14.5% of aromatic chemicals in crude oil [90]. The ability of *Prototheca zopfii* to break up crude oil was proven in laboratory experiments.

#### 3.1.3. Phenolic Compounds

Microalgae species are investigated for their ability to degrade phenolic compounds due to their inducible intracellular enzymes, such as polyphenol oxidase and laccase [86]. When it comes to biodegradation, the laccase enzymes from *Chlamydomonas moewusii* are primarily responsible [91]. The microorganisms *Pseudochlorococcum* sp., *Chlorella* sp., and *Chlamydomonas* sp. are all effective at treating phenol-contaminated wastewater [92]. Additionally, it has been demonstrated that the green microalga *Monoraphidium braunii* is a good species for producing monoglucoside from bisphenol [93]. The green microalgae *Chlorella* sp. and *S. obliquus* showed biodegradation capacity in the presence of several phenolic compounds that are identified as priority pollutants by the United States Environmental Protection Agency (EPA) [94]. Moreover, the phenol-resistant microalgae *Ankistrodesmus braunii* and *Scenedesmus quadricauda* were found to be highly efficient at degrading a variety of phenols, including catechol, hydroxytyrosol, p-hydroxy, tyrosol, benzoic acid, ferulic acid, synaptic acid, caffeic acid, and vanillic acid by approximately 70% in 400 mg/mL [68]. Previously, it was revealed that some microalgae possess some of the greatest observed activity for converting naphthalene to 1-naphthol [95]. Additionally, the marine microalga *Dunaliella* sp. was found to degrade dimethyl phthalate (DMP) more rapidly than *C. pyrenoidosa* [96].

#### 3.1.4. Pesticides and Herbicides

Recently, low-cost algal-based systems for the treatment of wastewater or effluents from agrochemical industries have been recommended for commercial application [97]. Numerous microalgae can assimilate various pesticides through biosorption and bioaccumulation, depending on their lipid content, strain, and the chemical structure of the pesticide [67,98]. Moreover, microalgae utilize pesticides and cyanide as their carbon and nitrogen sources [99].

Numerous microalgae can assimilate hazardous organic substances, including petroleum hydrocarbons, through the secretion of degrading enzymes [90,100,101]. The combination of *C. vulgaris* and biosurfactants is a superior approach for pretreatment of wastewater, particularly for nutrient removal from petrochemical wastewaters [102]. Many green algae species have been shown to have a high potential for degrading organic xenobiotics, such as pesticides, medicines, and herbicides, which have become a major issue in aquatic environments [103]. Diazinon is a toxic insecticide with a deleterious effect on various biota at high concentrations; it can be digested by *Chlorella vulgaris* and then transformed into a less poisonous metabolite [104]. It has been observed that the microalgae *Scenedesmus* sp. and *Chlorococcum* sp. convert endosulfan (a cyclodiene pesticide) to endosulfan sulfate [105]. *C. vulgaris* and *S. bijugatus* were reported to obtain phosphorus from an organophosphorus pesticide [106]. Pesticide bioremediation is dependent on the cytochrome P450 monooxygenase enzyme superfamily [107].

Numerous green microalgae species, notably *Chlamydomonas reinhardtii*, are used to bioremediate water systems contaminated with the herbicide prometryne [103]. Certain microalgae may be capable of rapidly degrading the pesticide fluroxypyr [108]. *C. vulgaris* was found to accumulate the triazine group of herbicides within 12 h [109]. Cytochrome P450 is a dealkylating enzyme activated when herbicides are degraded by algae [110]. The remediation process relies on monooxygenase enzymes generated in *C. fusca* and *C. sorokiniana* by the pro-herbicide metflurazon [111].

Nitroaromatic substances, such as trinitrotoluene (TNT), and a resistant xenobiotic employed as an explosive can be cracked by algae-secreted nitrate reductases [112].

Recently, it was demonstrated that the immobilization technique is superior to free cells in the biodegradation of various pollutants in wastewater. This may be attributed to a context of high population density with a low volume; immobilization reduced substrate inhibition and toxicity to microorganisms due to diffusional constraints, and it is a reusable system that reduces overall costs and enables cell storage for extended periods without impairing degradability [113]. Immobilization has been used in a wide variety of biotechnological applications for more than 40 years. A technique using Ca-alginate-immobilized *C. vulgaris* was reported to reduce the concentration of organic matter from industrial wastewater after 12 h [114]. It rapidly degraded tributyltin (TBT), an active ingredient in biocides, within one day. At a high TBT concentration, alginate-immobilized *C. vulgaris* detoxified TBT into DBT and MBT in a six-cycle period [115]. Additionally, *Chlorella emersonii* immobilized in alginate had a greater capability of degrading tri-, di-, and monobutyltin chlorides than free cells in aquatic solutions, and biocide buildup was also reduced [116]. According to [74,115], immobilized *Sc. quadricauda* in alginate is a viable alternative approach for dye degradation.

Algal–bacterial symbiosis is a costless treatment method for removal of organic pollutants from wastewater. Microalgae enhances bacterial biodegradation by providing the necessary oxygen, a crucial electron acceptor, for the aerobic bacterial degradation of organic contaminants. On the other hand, bacteria provide the CO_2_ required for microalgal photosynthesis [117]. This technique can significantly enhance the removal rate of pollutants, increase algal biomass and lipid productivity, and decrease the cost of microalgal harvest, making it a good prospect for large-scale practical application [118].

The microalgal *Sc. obliquus* bacterial consortium was found to degrade oil waste by 84.2% [119]. The *C. sorokiniana*–*Pseudomonas migulae* algal–bacterial combination was found to decompose roughly 350 ± 150 mgL^−1^ of phenanthrene from tetradecane or silicone oil without any external oxygen input in autotrophic settings [120]. Moreover, the consortium of *C. vulgaris* and *Coenochloris pyrenoidosa* was found to uptake 50 mg pentachlorophenol (PCP) L^−1^ under light conditions for five days [121].

In general, *Chlorella* spp. and *Scenedesmus* spp. are promising species for phycoremediation; *Scenedesmus* spp. was found to degrade contaminants more rapidly than other species [94].

### 3.2. Bioremediation of Organic Pollutants by Cyanobacteria

Cyanobacteria (blue-green algae) are prokaryotic microorganisms that thrive in a variety of conditions and have the potential to fix atmospheric nitrogen and carbon, hence increasing water fertility [122]. The biomass of the majority of cyanobacterial species is considered one of the most useful bioaccumulators due to their widespread distribution, the availability of low-cost cultivation technology, their adaptable metabolism, and their high absorption capacity [123,124]. Cyanobacterial species can degrade aromatic hydrocarbons and xenobiotics to less toxic or non-toxic components and utilize them as nutrition (Table 2). They release various enzymes, including laccase, azo reductase, and polyphenol oxidase, which are responsible for the remediation of various contaminants and modification of their metabolic processes [125]. Laccase and polyphenol oxidase of the cyanobacterium *Phormidium valderianum* have been identified as the enzymes responsible for most phenol biodegradation [126].

Thus, wastewater can be used to cultivate cyanobacteria species, increasing their production while still containing the biogenic components required for their growth [131]. *Westiellopsis* sp., *Spirulina* sp., and *Oscillatoria* sp. are the most commonly used cyanobacteria for industrial wastewater treatment [132,133,134,135].

#### 3.2.1. Dyes

*Hydrocoleum oligotrichum*, *Oscillatoria oligotrichum*, *O. limnetica*, and *Spirulina* spp. have been associated with the biodegradation of basic fuchsin, with rates varying according to the dye concentration and algal species [14]. *Phormidium animale* was found to be an efficient and safe but expensive method for degrading Remazol Black B (RBB) dye from wastewater [128]. *Arthrospira platensis* had a high capacity for removing reactive red 120 (RR-120) from aqueous solutions (482.2 mg g^−1^) [136]. Additionally, it was determined that *Nostoc linckia* is an appropriate cyanobacterium species for treating dyes in industrial wastewater [44].

*Anabaena flos-aquae* (UTCC64), *Synechococcus* sp. (PCC7942), and *P. autumnale* (UTEX1580) were found to have a high biodegradation efficacy toward three different dyes: Remazol Brilliant Blue R (RBBR), sulfur black, and indigo [137]. *Nostoc*
*linckia* degraded azo dye by 81.97% in seven days [66]. *Spirogyra rhizopus* was highly efficient at degrading acid blue [138]. After 26 days of treatment, *Gloeocapsa pleurocapsoides* and *P. ceylanicum* were found to decolorize FF Sky Blue and Acid Red 97 dyes by 80% [127].

*Nostoc linckia* HA 46 on calcium alginate was shown to effectively degrade toxic reactive red 198 dye by 94% at pH 2.0 and 30 °C in an aqueous solution containing calcium alginate (RSM) [44].

#### 3.2.2. Organic Hydrocarbon

The organic compounds di-n-butyl phthalate (DBP), diethyl phthalate (DEP), and dimethyl phthalate (DMP) are found in abundance in the environment. In comparison to *Microcystis aeruginosa* (Kütz.) (strain 2396, and SM), *Anabaena flos-aquae* from freshwater has the highest uptake rate for these compounds [137]. Pyrene is the first substance to contain PAHs with a high molecular weight. After 30 days of incubation, *Oscillatoria* sp. had a 95% greater rate of pyrene bioremediation than the green microalga *Chlorella* sp. (78.71%) [63].

Cyanobacteria species are capable of degrading petroleum and its derivatives [138]. *Anabaena* sp., *Aphanothece conferta*, *Phormidium* sp., *Nostoc* sp., and *Synechocystis aquatilis* degraded a variety of petroleum hydrocarbons, depending on the cyanobacteria species used and the chemical structure of the hydrocarbon compounds [139]. Cyanophyta species are critical in the decontamination of oil substances from waste and contribute to the hydrocarbon-degradation process [140]. Due to their efficacy in removing crude oil, mixed cultures, or individual cyanobacteria species, such as *O. salina, Plectonema terebrans* and *Aphanocapsa* sp., are used to alleviate oil pollution [141]. Immobilized *Phormidium animale* mat was found to degrade crude oil. *Ph. animale* acted exclusively as a matrix in this case, and the other microorganisms associated with this mat were the primary degraders [142]. It was reported that *Phormidium* sp. immobilized on synthetic capron fibers was an effective system for removing a mixture of phenols and oil spills [143]. When grown in light conditions, *A. variabilis* degraded 100% of o-nitrophenol (ONP), although this rate decreased in the dark [117,129].

Numerous cyanobacteria species are capable of degrading naphthalene (PAH) to four major compounds, including 1-naphthol, cis-naphthalene dihydrodiol, 4-hydroxy-4-tetralone, and trans-naphthalene dihydrodiol, at non-toxic concentrations, as illustrated in Figure 2 [95]. Under heterotrophic conditions, *N. punctiforme* and *Arthrospira platensis* were found to be capable of degrading crude oil and transferring aliphatic compounds to aromatic ones [82,144]. *Oscillatoria* spp. Was shown to be capable of degrading naphthalene, a major component of crude oil, and biphenyl [95].

Cyanobacteria polysaccharides play an important role in emulsifying oil and converting it into small droplets that are easily attacked during the heterotrophic process [145]. Cyanobacteria can degrade oil components [146]. According to Abed [138], aerobic heterotrophic bacteria–cyanobacteria consortia were found to be extremely beneficial for biodegradation in an oil-polluted area.

#### 3.2.3. Phenolic Compounds

Cyanobacterial species were able to tolerate and remove phenol from different system within days; polyphenol oxidase and laccase enzymes were first isolated from the marine cyanobacterium *Phormidium valderianum* [126]. *Lyngba lagerlerimi*, *N. linkia*, and *O. rubescens* can effectively remove a wide variety of phenolic pollutants (Table 1) [147]. Klekner and Kosaric [94] reported that *Spirulina maxima* could degrade a variety of phenolic compounds that are classified as priority pollutants in the United States (by the Environmental Protection Agency) as shown in Figure 3. The freshwater cyanobacteria *Anabaena cyalindrica* and *Phormidum foveolarum* were the first algal species identified as capable of bioremediating phenolic compounds without the production of metabolites or ring cleavage [148].

#### 3.2.4. Pesticides and Herbicides

Numerous cyanobacteria species were found to be capable of bioremediating the degradation of fenamiphos, a toxic pesticide [149]. *Aulosira fertilissima* ARM 68 used pesticides such as dichlorvos, quinalphos, malathion, monocrotophos, and phosphamidon as additional phosphorus sources when inorganic phosphate was available. In the absence of inorganic phosphate, they became the sole source because these pesticides acted as inducers for acid-phosphatase activity [150]. *Spirulina* spp. can metabolize glyphosate, a synthetic herbicide whose uptake rate is dependent on the cell’s phosphorus availability [151]. Within a day, *A. cyalindrica* and *M. aeruginosa* were found to degrade the toxic phenylurea herbicides [87,128]. *N. ellipsosporum* and *Anabaena* sp. converted lindane (a highly chlorinated aliphatic pesticide) into simple compounds [152]. *Oscillatoria quadripunctulata* absorbed dissolved solids from petrochemical waste, such as aromatic compounds, phenols, sulfides, and biocides, and subsequently reduced the concentration of total dissolved salts by 40% [130]. Water fertility may be improved by cyanobacterial nitrogen and carbon fixation [122,124]. Cyanobacterial and microalgal biomass is one of the most useful bioaccumulators due to its global distribution, low cost, and high absorption capacity [123]. Cyanobacteria feed on xenobiotics and aromatic hydrocarbons and break them down. In addition to laccase, cyanobacterial enzymes include azo reductase and polyphenol oxidase. Most phenol biodegradation occurs in Phormidium valderianum laccase and polyphenol oxidase [126]. Therefore, growing cyanobacteria in wastewater increases production while retaining biogenic components [131]. *Westiellopsis* sp., *Spirulina* sp., and *Oscillatoria* sp. The biodegradation of basic fuchsin has been linked to *Hydrocoleum oligotrichum*, *Oscillatoria oligotrichum*, *O*. *limnetica*, and *Spirulina* spp. [63]. The degradation of Remazol Black B (RBB) dye from wastewater was shown to be efficient and safe but costly [53]. *Arthrospira platensis* was found to be capable of removing RR-120 from aqueous solutions (482.2 mg g^−1^) [77]. *Nostoc linckia* is also suitable for treating colors in industrial effluent [78]. Biodegradation efficacy of three dyes—Remazol Brilliant Blue R (RBBR), sulfur black, and indigo—was determined in three different blue-green algae: *Anabaena flos-aquae* (UTCC64), *Synechococcus* sp. (PCC7942), and *P. autumnale* (UTEX1580). On average, azo-dye degradation took 7 days for *Nostoc linckia* [6]. Acid blue was efficiently degraded by *Spirogyra Rhizopus* [80]. *G. pleurocapsoides* and *P. ceylanicum* decolorized FF Sky Blue and Acid Red 97 dyes by 80% after 26 days [129]. At pH 2.0 and 30 °C, *Nostoc linckia* HA 46 on calcium alginate degraded the hazardous reactive red 198 dye by 94% [76]. Di-n-butyl phthalate (DBP), diethyl phthalate (DEP), and dimethyl phthalate (DMP) are abundant in nature. *Anabaena flos-aquae* had the highest absorption rate for these chemicals compared to *Microcystis aeruginosa* (Kütz.). *Oscillatoria* sp. had a 95% greater rate of pyrene bioremediation than that of the green microalga Chlorella sp. (78.71%) [31]. Cyanobacteria species are capable of degrading petroleum and its derivatives [32,138]. *Synechocystis aquatilis* and *Anabaena* sp. degraded various petroleum hydrocarbons [3]. Hydrocarbon breakdown and oil cleaning require cyanophyta [78]. *Aphanocapsa* sp. or mixed cultures of *O. salina* and *Plectonema terebrans* are used to minimize oil contamination [36]. *Phormidium animale* mat degraded crude oil [79]. This alga immobilized on synthetic capron fibers removed phenols and oil spills [80]. The primary PAH molecules are 1-naphthol, cis- and trans-naphthalene dihydrodiol, 4-hydroxy-4-tetralone, and 4-hydroxy-4-tetralone degraded by a variety of cyanobacteria, as shown in Figure 2 [77]. These organisms can degrade aliphatic chemicals and shift them to aromatic molecules [36,77]. *Oscillatoria* spp. degraded naphthalene, a component of crude oil, and biphenyl [3]. In oil-polluted areas, Abed [138] found that aerobic heterotrophic bacterial–cyanobacterial consortia were particularly advantageous. *Lyngba lagerlerimi*, *N. linkia*, and *O. rubescens* were found to effectively remove a wide variety of phenolic pollutants (Table 1) [147]. Klekner and Kosaric [94] claim *Spirulina maxima* can degrade phenolic compounds, which are considered pollutants in USA (by the Environmental Protection Agency). Hydrophobic petroleum molecules can resist microbial breakdown and survive in various environments (low water solubility). Blue-green algae are a common phototrophics and may grow in a variety of environments and generate nitrogen-fixing blooms [10]. The presence of cyanobacteria in polluted water has prompted toxin-tolerance studies [32,33,53].

Genetic engineering (GE) plays an important role in the bioremediation process to modify microalgae and cyanobacteria. GE increases the number of active groups of the algal cell wall to stimulate polysaccharide secretion to attract pollutants [50]. Engineered organisms can degrade a variety of organic pollutants, including PAHs, explosives, and aromatic and xenobiotic compounds. Additionally, modified algae are resistant to various pollutants and accelerate the degradation of pollutants under a variety of conditions [38]. Kuritz and Peter [152] previously demonstrated the efficacy of GE in enhancing the biodegradation ability of the cyanobacterium *Anabaena* sp. They revealed that *Anabaena* sp. containing the pRL634 gene degraded lindane more efficiently than the wild-type species. Regrettably, there are some barriers to applying GE, such as regulatory constraints and ecological concerns. To adapt genetic engineering for biodegradation applications, four major techniques should be included: (1) modification of enzyme specificity and affinity; (2) development and monitoring of bioprocesses; (3) regulation of the biodegradation pathway; and (4) application of bioreporter sensors and analysis endpoints to reduce pollutant toxicity [153].

## 4. Advantages of Phycoremediation Treatment

Many conventional treatment processes have been applied for wastewater treatment, such as precipitation; adsorption; coagulation; and advanced oxidation processes, including ozonation, UV, UV/H_2_O_2_, and photo–Fenton reaction. Although many of these techniques have been found to be efficient in the removal of various organic pollutants from ecosystem, they are associated with high costs and the production of harmful/toxic secondary products (Table 3) [154,155]. On the other hand, activated sludge is one of the most widely used techniques under aerobic circumstances, making use of a dense microbial culture in suspension to biodegrade organic material [156]. This method is characterized by the removal of soluble and suspended organic matter from wastewater but produces bulk waste and sludge [157]. Recently, the photo-oxidation of organic pollutants using TiO_2_ nanoparticles has attracted a lot of interest due to its effective role in oxidizing and mineralizing a broad range of hazardous organic pollutants [154].

Phycoremediation is considered an efficient approach that is environmentally safe and sustainable environmental treatment that does not generate large amounts of secondary waste (sludge). Compared to various physical and chemical technologies that are typically expensive and ineffective, phycoremediation is a cost-effective technique with limited versatility that does not address carbon sequestration, sludge production, or the prevention of the formation of carcinogenic intermediates (Figure 4). The investment cost of biological processes is 5–20 times less than that of conventional chemical procedures. In comparison, the running cost is 3–10 times less than that of conventional procedures [158]. Additionally, phycoremediation can be viewed as a form of permanent bioremediation, as it may result in the complete mineralization of pollutants, as well as a blue and circular economy [9,159].

Microalgae and cyanobacteria are known as promising candidates for the biodegradation microorganisms for a variety of pollutants [70] in comparison to bacteria and fungi, which require carbon input, energy, nutritional sources, and other supplements to remove pollutants, with fixed carbon eventually entering the atmospheric carbon pool, which is increasing alarmingly as a result of the use of fossil fuels for a variety of human activities [65,110]. Another economic benefit of phycoremediation is the high production of algal biomass, which is facilitated by organic pollutants, resulting in high pollutant absorption and accumulation. Additionally, it was discovered that the biodegradation efficiency of green algae *Chlorella* and *Scenedesmus* is greater than that of several bacterial strains, including *Rhodococcus* sp. [160]. The desulfonation of naphthalene monosulfonic acids by *S. obliquus* is more rapid than same the process by some bacteria [161,162]. Mixotrophy microalgae and cyanobacteria have distinct competitive advantages over bacteria and fungi. The biodegradation of organic pollutants, as metabolic and genomic information, aids not only in the identification and selection of the appropriate species capable of degrading organic pollutants but also in monitoring remediation under field conditions [110]. Bioremediation with plants, called phytoremediation, is the most attractive method for converting diverse contaminants to innocuous compounds; however, many pollutants cannot be handled with plants [163]. Microalgae are more efficient than plants due to their rapid development rate and low cultivation requirements, as they do not require land [163] and are advocated for phycoremediation of shallow contaminated areas (Figure 4).

Dyes discharged as effluent from companies are treated using biological, chemical, and physical processes. Adsorption is the most common and cost-effective method for the removal of dyes. Several studies have used waste materials as dye adsorbents [159]. Each remediation technique has benefits and drawbacks that must be assessed individually. The main disadvantages of dredging and capping are their significant environmental impact and high investment requirements. As biological systems, bioremediation approaches have low predictability and sometimes long degradation times, necessitating extensive monitoring. Additionally, many biodegradation experiments are conducted in laboratories, where factors influencing these systems differ from those in the field [9]. In the future, scientists hope to better understand how plants and microorganisms interact to achieve environmental remediation [164].

## 5. Conclusions and Future Perspectives

Due to the increasing use of hydrocarbon waste and other organic pollutants in different systems, they have become one of the world’s most serious environmental challenges. Bioremediation of various organic pollutants by microalgae and cyanobacteria is a sustainable and environmentally acceptable green technology for treatment of polluted water that has a lesser environmental impact than other microorganisms and traditional methods. Furthermore, these contaminants promote the growth of algal biomass, which can be exploited in a variety of ways in the future. To tackle this difficulty, future attempts will be made to screen for more algae strains for phycoremediation, as well as genetic engineering to improve algal biodegradation capability and tolerance to various organic contaminants.

Moreover, the relocation of novel pollutant-degrading bacterial genes into algae will be applied to accelerate the degradation rate of a variety of organic contaminants. To accelerate the phycoremediation of organic contaminants and reduce decontamination time, the physicochemical characteristics of aquatic systems, such as temperature, pH, and nutrient availability, must be altered. Effective design of the growing system is critical to maximize growth rates and lower costs.

## Figures and Tables

**Figure 1 molecules-27-01141-f001:**
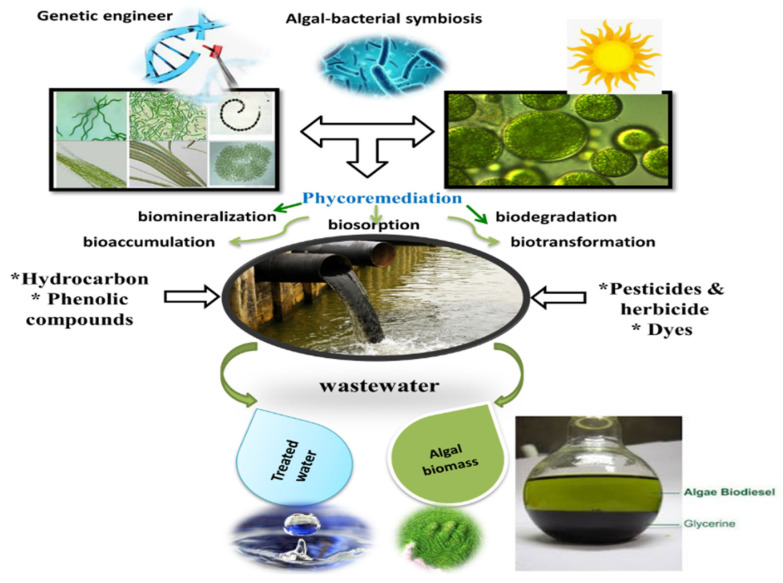
Phycoremediation of organic pollutants by microalgae, cyanobacteria and biomass utilization.

**Figure 2 molecules-27-01141-f002:**
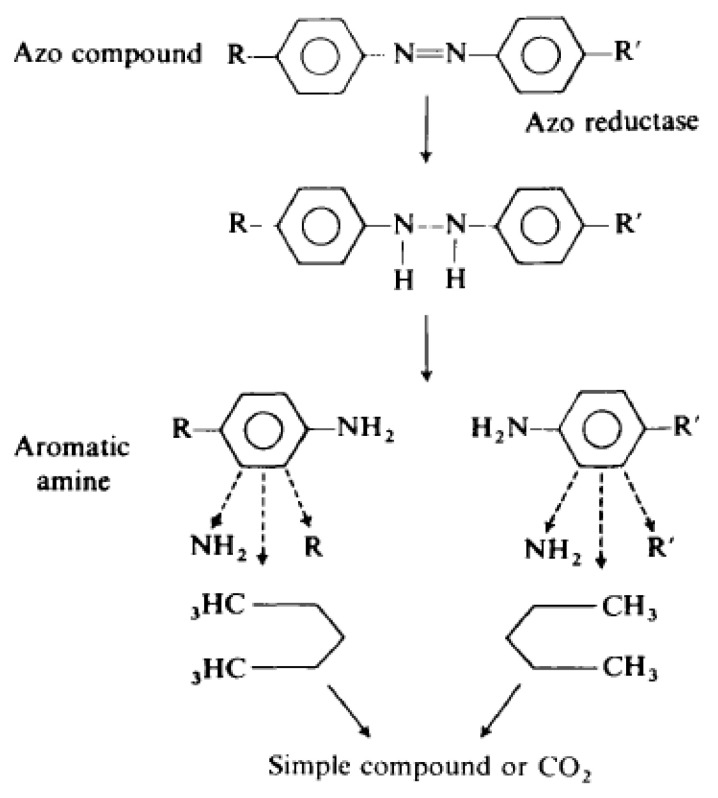
Degradation mechanism of azo dyes by microalgae. Reprinted with permission from [73]. Copyright 1992 Elsevier.

**Figure 3 molecules-27-01141-f003:**
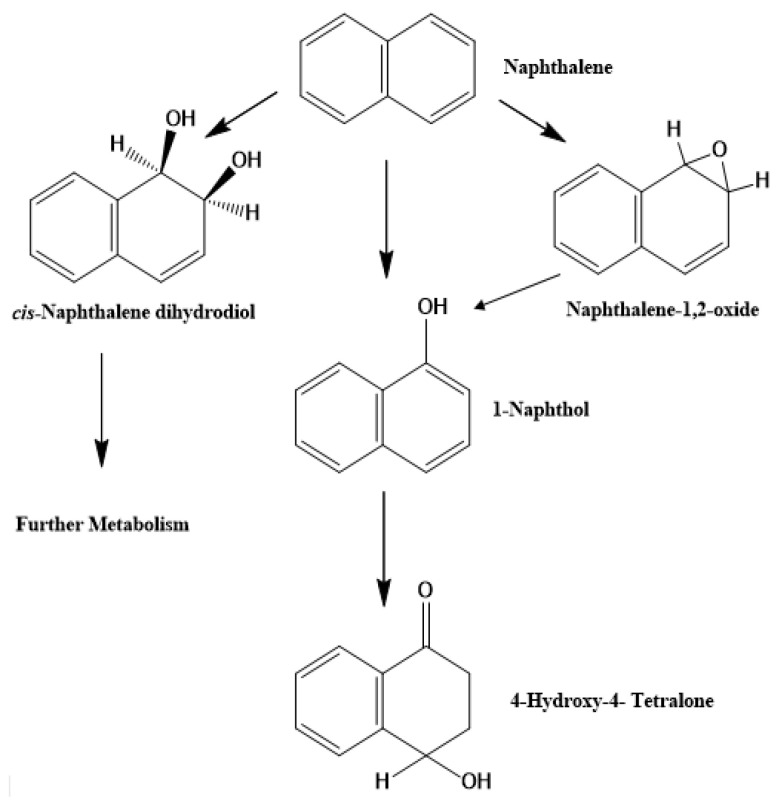
Cyanobacterial transformation mechanisms of naphthalene. Reproduced with permission from [94]. Copyright 1992 Informa UK Limited.

**Figure 4 molecules-27-01141-f004:**
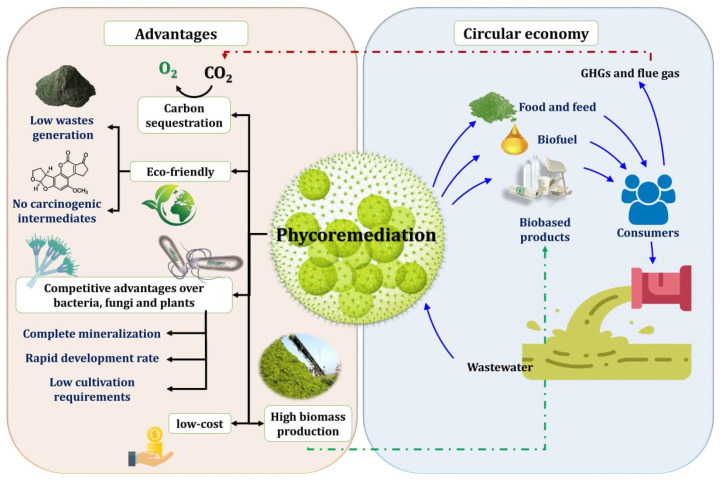
The advantages and effective roles of algae in phycoremediation of wastewater.

**Table 1 molecules-27-01141-t001:** The removal efficiency of organic pollutants using microalgae on a laboratory scale.

Pollutants	Algae Species	Organic Pollutants	Degradation Rate %	References
Dyes	*Chlorella* spp.	Pyrene	78.71	[63]
	*Chlorella vulgaris*	Azo dye	≥90	[64]
	*Sc. Bijugatus*	Tartrazine	57	[65]
	*Volvox aureus*	Basic cationic (10 ppm)	82	[66]
Hydrocarbon	*Chlorella* spp.	Pyrene	78.71	[63]
	*Sc. Obliquus*-bacterial consortium	Oil wastes	84.2	[67]
Phenols	*Ankistrodesmus braunii* and *Sc. Quadricauda*	phenols	70	[68]
Pesticides	*Nannochloris oculata*	Lindane (0.1 mg/L)	73	[68]
	*Chlamydomonas reinhardtii*	Isoproturon (50 µg/L)	15.1	[69]

**Table 2 molecules-27-01141-t002:** The removal efficiency of organic pollutants using cyanobacteria on a laboratory scale.

Pollutants	Algae Species	Organic Pollutants	Degradation Rate %	References
Dyes	*Nostoc linckia* HA 46	Toxic reactive red 198 dye	94	[44]
	*N. muscorum*	Tartrazine	70	[65]
	*Nostoc linckia*	Azo dye	81.97	[66]
	*Oscillatoria rubescens*	Basic Fuchsin (5 ppm)	94	[66]
	*Phormidium ceylanicum*	Acid Red 97	89	[127]
	*Ph. animale*	Remazol Black B (RBB)	99.66	[128]
	*Chroococcus minutus*	Amido Black 10B (100 mg L^−1^)	55	[127]
	*Gloeocapsa pleurocapsoides*	FF Sky Blue (100 mg L^−1^)	90	[127]
Hydrocarbon	*Prototheca zopfii*	Saturated aliphatic hydrocarbons	49 ±11	[90]
	*Prototheca zopfii*	Aromatic compounds	26.5 ± 14.5	[90]
	*Oscillatoria* sp.	Pyrene	95	[63]
Phenol	*Anabeana variabilis*	O-nitrophenol (ONP)	100	[129]
Pesticides & herbicides	*Oscillatoria quadripunctulata*	Biocides	40	[130]

**Table 3 molecules-27-01141-t003:** Comparison between removal techniques utilized for wastewater treatment.

Disadvantages	Advantages	Process
Phycoremediation	Ecofriendly, low cost	Not effective for some pollutants Limiting pH tolerance
Fungi and bacteria	Cost-effective Low energy requirements	Sludge production
Activated sludge	Effective method for removal of soluble and suspended organic material	Production of toxic secondary pollutants
Chemical precipitation	Adapted to high levels of pollution Simple process	Production of sludge Chemical consumption
UV/H_2_O_2_	Effective method for mineralization and oxidation of most organic pollutants	Costly technique Less effective
Electrochemical oxidation	Improves biodegradability Does not require chemicals or high temperatures	Low reaction rates and selectivity High initial cost of the equipment Formation of sludge
Ozonation	Applicable for a wide range of pollutants	High cost Complex technology
TiO_2_	Effective method for mineralization and oxidation of most organic pollutants	Formation a harmful byproduct

## Data Availability

Not applicable.

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
