# Peer review of "A Review of Microalgae- and Cyanobacteria-Based Biodegradation of Organic Pollutants"

_molecules, 2022, doi:10.3390/molecules27031141_

Round 1

Reviewer 1 Report

I think a review must contain two parts. The first part is the evidence of extensive reading, which is clearly shown in this review. It is evident that the references are numerous.

The second part is critical thinking. I would expect the reviewer to express his point of view on the articles that he read. I think this part is lacking in this review. This review consists mainly of a rephrasing of previous publications, without critical analysis by the authors.

The reviewer is on microalgae and cyanobacteria to degrade organic pollutants. I think the major question to answer is why did the microalgae and cyanobacteria degrade the organic pollutant? What is the benefit of doing so? Both of these groups are photoautotrophs, which means they are using CO2 as their carbon source and light as their energy source. They do not need the carbon from the organic pollutant. Therefore, why did they degrade the organic pollutant? I think this matter needs to be addressed in this review.

Lanes 106 - 112

The adsorption and absorption are ideal for removing pollutants by algae. But, it also means that the algal biomass needs to be removed from the aquatic environment. If they are left to die and degrade, the pollutant would be released back into the environment. I think this issue needs to be addressed. 

lane 109 - I do not think it is going to be cheap to remove the algae containing the pollutants from the environment. The algae might bloom and upon degradation, they would consume oxygen and this is not eco friendly.

lane 114- The algae would remove phosphorus and nitrate from the water column by consuming them. By doing so, they may grow rapidly and cause eutrophication  (as mentioned in line 125). The author implies that the algae might be able to deter eutrophication by consuming the nutrients. In reality, algae are the ones that can cause eutrophication while consuming the nutrients. I think this matter needs to be addressed.

Lanes 159- 165 - The algae can remove pollutants via biosorption and bioaccumulation. But the biomass of the algae must be removed from the water column, together with the contaminants. Otherwise, the algae would die and release the contaminants back to the environment. This issue was not addressed.

lanes- 199 -205

Why did the algae convert the compounds into CO2? The algae are photosynthetic,  not heterotopic organisms. I would take the statement with a pinch of salt.

Lines 211-219 - Again, why did the algae break the hydrocarbons?

Lines 220-223 - It is clearly stated that the algae are breaking down the hydrocarbon under heterotrophic conditions using oil waste as the cole carbon source? Why these species are capable of doing that?

Lane301-388 - What is the benefit of breaking these hydrocarbons towards cyanobacteria? Why did the cyanobacteria degrade the hydrocarbon?

lane 405 - This review only mentions the advantage of using cyanobacteria and microalgae to degrade hydrocarbon. How about the disadvantages? Some species like Anabaena (line 397 etc) are capable of producing toxins. The toxin would also create problems if the species are used. 

Lane 409 - 411. Alge does not produce sludge? How about the algal biomass? How to get rid of them? If not, wouldn't they become a problem too?

Lane 417 - 420 - If the biodegradation by algae would not release any carbon into the atmosphere, where would the carbon eventually end up at?

Author Response

Reviewer 1

Comments and Suggestions for Authors

  1. I think a review must contain two parts. The first part is the evidence of extensive reading, which is clearly shown in this review. It is evident that the references are numerous.

  • Response

Thank you for the respected reviewer for this comment about this point. We have considered this suggestion in the text and it is highlighted by different color in the text.

  1. The second part is critical thinking. I would expect the reviewer to express his point of view on the articles that he read.

  • Response Thank you for the respected reviewer for this comment. We have modified this part.

  1. I think this part is lacking in this review. This review consists mainly of a rephrasing of previous publications, without critical analysis by the authors.

  • Response

We believe that any article review research is just a collection of research that deals with an important research point. In this research, we address the extent of biodegradation based on cyanobacteria for organic pollutants. In addition, we have added some opinions and some new references with discussion and also, we added illustration to be clearer for the readers tio understand the main issue of the review.

  1. The reviewer is on microalgae and cyanobacteria to degrade organic pollutants.
    • I think the major question to answer is why did the microalgae and cyanobacteria degrade the organic pollutant?
    • What is the benefit of doing so? Both of these groups are photoautotrophs, which means they are using CO2 as their carbon source and light as their energy source.
    • They do not need the carbon from the organic pollutant. Therefore, why did they degrade the organic pollutant? I think this matter needs to be addressed in this review.

  • Response

Microalgae and cyanobacteria are used in phytoremediation to remove nutrients and toxins from wastewater and carbon dioxide from the air [1,2]. The photoautotrophic bacteria are desired since their use is eco-friendly and does not pollute the environment [3]. Many researchers use cyanobacteria in phycoremediation because their ability to develop quickly, can grow on non-arable land, their low water and land requirements as well as photoautotrophic microorganisms [4-7]. Blue Green algae can consume CO2 and convert to algal biomass and increase the levels of O2. These process can be used for producing biogas, biofuels and many useful production [7]. Furthermore, algae and cyanobacteria are capable of degrading pollutants by excretion of some enzymes which degrade the toxic compounds into simple nontoxic compounds, and this is illustrated in the text and in tables and figures.

  1. Lanes 106 - 112
    • The adsorption and absorption are ideal for removing pollutants by algae. But, it also means that the algal biomass needs to be removed from the aquatic environment. If they are left to die and degrade, the pollutant would be released back into the environment. I think this issue needs to be addressed.

  • Response: The authors thank the reviewer for his keen observation. We have addressed this point in the manuscript.

Biosorption is a series of activities that occur in the cell wall before anaerobic or aerobic biodegradation. It has good selectivity and efficiency (high performance and low cost). Biosorbents might be natural materials like marine algae or weeds or industrial wastes like activated sludge or fermentation wastes [8]. The best method for the elimination of biological sludge mechanically. The dangerous of biomass sludge were depend on the materials absorbed or adsorbed. Using biosorption to remove organic contaminants as pesticides, phenols were discussed in a review study by Aksu [9].

  1. lane 109 –
    • I do not think it is going to be cheap to remove the algae containing the pollutants from the environment. The algae might bloom and upon degradation, they would consume oxygen and this is not eco friendly.

  • Response

Removing dangerous and toxic pollutants from the environment, even if they are expensive, is better than leaving them and then consider eco-friendly. We consider it ecofriendly since it can not only adsorb pollutants but also it degrades the pollutants into simple compound such as azodyes. The algae and cyanobacteria can degrade it through excretion of specific enzymes such as azodyes reductase and convert dyes to simple nontoxic compounds such as NH­ and CO2.

  1. lane 114-
    • The algae would remove phosphorus and nitrate from the water column by consuming them. By doing so, they may grow rapidly and cause eutrophication  (as mentioned in line 125). The author implies that the algae might be able to deter eutrophication by consuming the nutrients. In reality, algae are the ones that can cause eutrophication while consuming the nutrients. I think this matter needs to be addressed.

  • Response: The authors thank the reviewer for his keen observation. We have addressed this issue as suggested

The literature review about the efficiency of phycoremediation to reduce the nutrient levels from eutrophic lakes are not too much. On the other hand, Razak [10] discusses the promise of algal-based approaches in resolving lake eutrophication. They reported that, wastewater is the primary focus of most phycoremediation studies Using microalgae to reduce fertilizer, pollution, and heavy metal levels in lakes is a surefire way to improve water quality. Lake phycoremediation has received little attention from researchers. Therefore, algae must have more studied for their ecological and physiological features especially in eutrophication. Phycoremediation's effectiveness must be studied in natural settings such as lakes and ponds. Bioremediation uses organisms like bacteria, fungi, and algae that can break down toxins metabolically. Nutrient enrichment promotes the growth of naturally occurring algae in all water bodies. These indigenous bacteria work similarly to those utilized in bioremediation processes when algae release oxygen into the air. Photosynthetic algae provide clean oxygen into the atmosphere, improving water quality, marine life, and biodiversity [11,12]. Phycoremediation preserves the lake's natural food chain, making it the most efficient nutrient removal method. Algae and natural microbes work together to clean water. Algae use CO2 and nutrients to generate O2 [13,14] and no trash is created.

  1. Lanes 159- 165 –
    • The algae can remove pollutants via biosorption and bioaccumulation. But the biomass of the algae must be removed from the water column, together with the contaminants. Otherwise, the algae would die and release the contaminants back to the environment. This issue was not addressed.

  • Response The authors thank the reviewer for his keen observation. As we mentioned above the algae and cyanobacteria not only bioremediate pollution via biosorption or bioaccumulation but also through biodegradation.

Among the harmful effects of pollution, eutrophication is the most common. High amounts of organic materials and decomposing organisms deplete oxygen in the water and kill other species. The nutrients (NH4+, NO3-, PO43-) in secondary effluents are the principal source of eutrophication in natural waters. These nutrients degrade water quality and harm the aquatic ecosystem [2,15]. So, before to release into waterbodies, wastewater must be appropriately treated. Activated sludge (for nitrogen and phosphorous removal), electro flocculation, membrane filtration, electro-kinetic coagulation, electro-chemical destruction, ion exchange, katox treatment (catalytic oxidation) and disinfection (by ozonation, chlorination, ultraviolet light) are some of the unit processes used to remove nutrients from wastewater [2].

  1. lanes- 199 -205
    • Why did the algae convert the compounds into CO2? The algae are photosynthetic, not heterotopic organisms. I would take the statement with a pinch of salt.

  • Response: Thank you for the respected reviewer for this comment. Since this a review article, we mentioned the results of the published papers which investigated the conversion of the toxic compounds to simple compound such as CO2. One of the authors of this review article (El-Sheekh) also published several papers about the biodegradation ability of algae and cyanobacterioa to pollutants through enzymatic machinery.
    • We mention according to (Jinqi and Houtian) the conversion of azo compounds into simpler aromatic amines or CO2, depend on the molecular structure of the dyes and the algal species used. Also, the aromatic amine is then subjected to further metabolism by algae.

The ability of Scenedesmus quadricauda to degrade Reactive Blue 19 and Remazol Brilliant Blue R dye (RBBR) in a variety of aquatic environments was discussed by [16]. It is well known that algae are photosynthetic and can occupied any environment. Their crucial roles in nutrient cycle and oxygen production are significant in many ecosystems. Biomass from microalgae has been utilized as an adsorbent. Scenedesmus quadricauda is a newcomer to biosorption research. They use light instead of carbon sources like bacteria and fungus, allowing them to survive in the lack of organic carbon. So, using metabolically active microalgal systems may be easier [16]. Recently, immobilized algae were used to solve this issue. Furthermore, C. ellipsoidea, C. kessleri, C. vulgaris, Sc. bijuga, Sc. bijugatus, and Sc. obliquus were able to decompose both mono- and di-azo Tartrazine [17].

  1. Lines 211-219 –
    • Again, why did the algae break the hydrocarbons?

  • Response

Through laboratory experiment, it has been proven that the used algae are able to break down hydrocarbon compounds, we added the following:

Some microalgae associated with crude oil may have remediation activity due to their ability to break down hydrocarbons into less complex chemicals without causing harm [18]. El-Sheekh [18] reported that, N. punctiforme and S. platensis can grow under heterotrophic condition using crude oil as a source of carbon (heterotrophic condition). Benzo[a] pyrene (BaP) is a carcinogenic polycyclic aromatic hydrocarbon (PAH) with a high molecular weight; it can be degraded by many microalgae species, including C. vulgaris, Sc. quadricauda, Sc. platydiscus, and Selenastrum capricornutum depending on the algae's density, cell wall composition, and oxidation-reduction enzymes involved in [19]. Subashchandrabose [19] confirmed that, microalga, Chlorella sp. Capable for degredation of pyrene through production of dihydrolipoamide acetyltransferase leading to remediating soils contaminated with pyrene. Selenastrum capricornutum has previously been shown to oxidize benzo[a]pyrene to sulfate ester and - and -glucoside conjugates using a dioxygenase mechanism [20-23]. Pyrene biodegradation occurs at a species-specific rate that is mostly dependent on the algal biomass concentration [22,23]. The authors reported that, Pyrene's growth-inhibiting effects on algae may be mediated through suppressing protein synthesis. Conversely, bacterial pyrene metabolites may enhance algal growth by stimulating DNA replication and protein synthesis. This research shows how algae and bacteria work together to degrade organic pollutants and reduce their toxicities to algae.

  1. Lines 220-223 –

It is clearly stated that the algae are breaking down the hydrocarbon under heterotrophic conditions using oil waste as the cole carbon source? Why these species are capable of doing that?

  • Response

Both C. vulgaris and Sc. obliquus have been discovered as potential biosystems for crude oil degradation due to their rapid growth rates and capacity to biodegrade oil under heterotrophic conditions using waste oil as the sole carbon source [24-26]. El-Sheekh [26] focused the ability of S. obliquus and C. vulgaris to grow under heterotrophic conditions using crude oil as sole carbon source. They proved the ability of both algae to degrade n-alkane and PAHs. This considered a talented technique and an economically feasible method which could be used at large scale. Ghasemi reported that, algae can promote pollutant degradation by either improving the microbial community's degradation capability or altering the pollutant directly. As a result of its tolerance to organic pollutants, Chlorella has colonized wastewater treatment systems. In addition to product selectivity, algae feature a recycling mechanism. Changing enzyme function by genetic engineering may produce useful enzymes. Previously, the green alga Prototheca zopfii was capable of degrading around 49 ± 11% of saturated aliphatic hydrocarbons and 26.5 ± 14.5% of aromatic chemicals in crude oil, respectively [27]. Through laboratory experiments that were done on Prototheca zopfii, it was proven its ability to break up crude oil.

  1. Lane301-388 –

What is the benefit of breaking these hydrocarbons towards cyanobacteria? Why did the cyanobacteria degrade the hydrocarbon?

  • Response

Water fertility may be improved by cyanobacterial nitrogen and carbon fixation [28]. Cyanobacterial biomass is one of the most useful bio-accumulators due to its global distribution, low cost, and high absorption capacity [29]. They feed on xenobiotics and aromatic hydrocarbons and break them down. In addition to laccase, cyanobacterial enzymes include azoreductase and polyphenol oxidase. Most phenol biodegradation occurs in Phormidium valderianum laccase and polyphenol oxidase [30]. So growing cyanobacteria in wastewater increases production while retaining biogenic components [31]. Westiellopsis sp., Spirulina sp., and Oscillatoria sp. Basic fuschin biodegradation has been linked to Hydrocoleum oligotrichum, Oscillatoria oligotrichum, O. limnetica, and Spirulina spp. [1]. The degradation of Remazol Black B (RBB) dye from wastewater was shown to be efficient, safe, but costly [2]. Arthrospira platensis could remove RR-120 from aqueous solutions (482.2 mg g-1) [3]. Nostoc linckia is also suitable for treating colors in industrial effluent [4]. Biodegradation efficacy of three dyes: Remazol Brilliant Blue R (RBBR), sulfur black, and indigo was determined in three different Blue green algae: Anabaena flosaquae (UTCC64), Synechococcus sp. (PCC7942), and P. autumnale (UTEX1580). On average, azo dye degradation took 7 days for Nostoc linckia [6]. Acid blue was efficiently degraded by Spirogyra Rhizopus [7]. G. pleurocapsoides and P. ceylanicum decolorized FF Sky Blue and Acid Red 97 dyes by 80% after 26 days [8]. At pH 2.0 and 30°C, Nostoc linckia HA 46 on calcium alginate degraded the hazardous reactive red 198 dye by 94% [1]. Di-n-butyl phthalate (DBP), diethyl phthalate (DEP), and dimethyl phthalate (DMP) are abundant in nature. Anabaena flos-aquae had the highest absorption rate for these chemicals compared to Microcystis aeruginosa (Kütz.). Oscillatoria sp. had a 95% greater rate of pyrene bioremediation than the green microalga Chlorella sp. (78.71%) [11]. Cyanobacteria species are capable of degrading petroleum and its derivatives [12,32]. Synechocystis aquatilis and Anabaena sp. degraded various petroleum hydrocarbons [3]. Hydrocarbon breakdown and oil cleaning require cyanophyta [4]. Aphanocapsa sp. or mixed cultures of O. salina and Plectonema terebrans are used to minimize oil contamination [5]. Phormidium animale mat degraded crude oil [6]. This alga immobilized on synthetic Capron fibers removed phenols and oil spills [7]. The primary PAH molecules are 1-naphthol, cis- and trans-naphthalene dihydrodiol, 4-hydroxy-4-tetralone, and 4-hydroxy-4-tetralone degraded by many of cyanobacteria, as shown in Fig. 2 [3]. These organisms can degrade aliphatic chemicals and shift them to aromatic molecules [4,5]. Oscillatoria spp. degraded naphthalene, a component of crude oil, and biphenyl [3]. In oil-polluted areas, Abed [32] found that aerobic heterotrophic bacterial–cyanobacterial consortia were particularly advantageous. Lyngba lagerlerimi, N. linkia, and O. rubescens effectively remove a wide variety of phenolic pollutants (Table 1) [33]. Klekner and Kosaric [34] claim Spirulina maxima can degrade phenolic compounds, which are considered as pollutants in USA (The Environmental Protection Agency). Hydrophobic petroleum molecules resist microbial breakdown and survive in environments (low water solubility). Blue-green algae are a common phototrophic and may grow in a variety of environments and generate nitrogen-fixing blooms [11]. The presence of cyanobacteria in polluted water has prompted toxin tolerance studies  [2,12,13].

  1. lane 405 –

This review only mentions the advantage of using cyanobacteria and microalgae to degrade hydrocarbon. How about the disadvantages? Some species like Anabaena (line 397 etc) are capable of producing toxins. The toxin would also create problems if the species are used.

  • Response

The same response to reviewer 3

Disadvantage of the phycoremediation treatment

Biotransformation can produce intermediates that are more hazardous than the original chemical, and some of these are nonbiodegradable and time consuming to break down [35]. There is a growing demand for eco-friendly solutions employing low coast because to the disadvantages of chemical treatments such as higher energy consumption, inadequate metal removal, and hazardous waste creation. Demand for microbial biomass for cleanup has risen in recent years. Phycoremediation is the use of macroalgae or microalgae to remove or bio-transform contaminants from wastewater and CO2. The major ingredients of algae are carbohydrates, protein, and phenolic chemicals, which contain metal bonding groups such as amines, carboxylates, and hydroxyls. Algal species are easy to grow, adapt, and manipulate in the lab [35-44].

  • We have answered this question from the reviewer 3.

  1. Lane 409 - 411.

Algae does not produce sludge? How about the algal biomass? How to get rid of them? If not, wouldn't they become a problem too?

  • Response

Thank you for the respected reviewer for this critical point and he is right in that, algae didn’t produce sludge, but it produce biomass and we have corrected it as he suggested.

Dyes discharged as effluent from companies are treated using biological, chemical, and physical processes. The removal of dyes is discussed. Adsorption is the most cost-effective way. The most common method is adsorption. Several studies have used waste materials as dye adsorbents [45]. Each remediation technique has benefits and drawbacks that must be assessed individually. The main disadvantages of dredging and capping are their significant environmental impact and high investment requirements. As biological systems, bioremediation approaches have low predictability and sometimes long degradation times, necessitating extensive monitoring. Also, many biodegradation experiments are conducted in laboratories, where factors influencing these systems differ from those in the field [46]. In the future, scientists hope to better understand how higher plants and the microorganisms interact to help with environmental remediation [47].

  1. Lane 417 - 420 –

If the biodegradation by algae would not release any carbon into the atmosphere, where would the carbon eventually end up at?

  • Response

Most of the algae that degrade pollutants, they transform it to simple compounds such as CO2 and this is illustrated in Fig. 1.

Reviewer 2 Report

The authors proposed a new bioremediation method based on algae's diverse functionalities. A greenway for cleansing wastewater is more ecologically friendly and environmentally sustainable than prior methods with other bacteria. However, there are some issues that need to be addressed. I suggest a major revision.

  1. The introduction part is too simple; some important background of the review is missing. The importance of introduction is very high, which is related to the idea and novelty of the whole review article, so the authors need to expound at a larger length.
  2. The context of the review is not time-sensitive, more recent research advances need to be reviewed. It is suggested to reference more research results in this field in recent years.
  3. What happened to the headline in Section 3.
  4. In section 3.2 “bioremediation of organic pollutants by cyanobacteria”, a list of relevant research advances is also required, as same as the lists in Table 1.
  5. In section 4 “Advantages of the phycoremediation treatment”, some visual diagrams or tables need to be provided to fully illustrate this section and make it more readable.

Author Response

Reviewer 2

The authors proposed a new bioremediation method based on algae's diverse functionalities. A greenway for cleansing wastewater is more ecologically friendly and environmentally sustainable than prior methods with other bacteria. However, there are some issues that need to be addressed. I suggest a major revision.

Thank you for the respected reviewer for this opinion about our work

  1. The introduction part is too simple; some important background of the review is missing. The importance of introduction is very high, which is related to the idea and novelty of the whole review article, so the authors need to expound at a larger length.

Response: Thank you for the respected reviewer. We added more important background and information

  1. The context of the review is not time-sensitive, more recent research advances need to be reviewed. It is suggested to reference more research results in this field in recent years.

Response: Thank you for the respected reviewer. We added a recent reference 2020, 2021 and 2022.

  1. What happened to the headline in Section 3.

Response: Thank you for this critical important point. The title is added (Bioremediation of organic pollutants)

  1. In section 3.2 “bioremediation of organic pollutants by cyanobacteria”, a list of relevant research advances is also required, as same as the lists in Table 1.

Response: Table 2 (bioremediation of organic pollutants by cyanobacteria) was separated from Table 1 as suggested by the respected reviewer.

  1. In section 4 “Advantages of the phycoremediation treatment”, some visual diagrams or tables need to be provided to fully illustrate this section and make it more readable.

Response: Thank you for this suggestion. Visual diagram is constructed and included

Reviewer 3 Report

The authors present a review of the use and applications of cyanobacteria and microalgae in wastewater treatment.

I would go directly so that the authors can understand why I would recommend a thorough revision and rewriting of most of the paper to be accepted:

Line 70: Please elaborate more on this sentence. It is not easy to understand as it is: “The adsorption method for the decolonization process was easy operation 70 and low cost [15]. »

Line 82 : « The toxicity of organometallic compounds is little understood, de- 82 spite being the most deadly substances on the market. » Reference here, please.

Line 92-95 : were being the most widely used pesticides and herbicides in agriculture (rewrite please to give clear sense)

Line 97 : use synonym for bad : Having them in water is bad for ecosystems (harmful ??? toxic ???)

Line 118-123 : rewrite this sentence : These findings will help us better understand how phytoremediation can be achieved 118 using cyanobacteria. Using Cyanobacteria as a bioremediation tool in recycling. Also, with 119 the interest of elucidating the nature of this particular algal assemblage. To improve agri- 120 cultural processes, cyanobacteria can be used because of their high activity in photosyn- 121 thesis and the ability to separate the bioactive material from these algal groups Microalgae 122 species for phytoremediation of organic contaminants in aquatic settings are discussed by 123 many authors [26,27].

Line 169-175: References 39,41 and 42 doesn’t seem to outline specifically the subject of biosurfactants from cyanobacteria and algae. I would recommend other references related to the production of such biomolecules specifficaly and making a mention of the types of molecules (biosurfactants that are produced). Besides, biosurfac- 169 tant-producing algae may significantly enhance the biodegradation of hydrocarbon-pol- 170 luted positions [39]. These biosurfactants are amphiphiles, thereby increasing the solubil- 171 ity of hydrophobic pollutants in water to accelerate hydrocarbon bioremediation [41]. PCB 172 biodegradation by algae is dependent on their physicochemical and physiological the con- 173 centration and hydrophobicity of PCBs, the algae's biomass and physiological capacities, 174 as well as the algae's membrane permeability [42].

Line 175-176: Where is the 3 Title???

Line 185: that can (be used to) manufacture biodiesel [46,47].

Line 210-211: Give sense to this sentence please: “Nitzschia sp. bioremediated phenanthrene (PHE) and fluoranthene (FLA) more rapidly 210 than Skeletonema costatum [61].”

Line 217: involved ?? erase in

Line 228-235: The green microalgae Chlorella sp. 228 and Sc. obliquus exhibited biodegradation ability associated with many phenolic com- 229 pounds, which are listed by the U.S. Environmental Protection Agency “EPA” as priority 230 pollutants [71]. Pseudochlorococcum sp., Chlorella sp., and Chlamydomonas sp. are effectively 231 utilized to treat phenol-contaminated wastewater [69].

And repeat?????

The green microalgae Chlorella sp. 232 and S. obliquus showed biodegradation capacity in the presence of several phenolic com- 233 pounds that are identified as priority pollutants by the United States Environmental Protection Agency "EPA" [71].

Line 277-300: A comprehensive table including type of algae, type of chemical, degradation rate, and reference for all the chemicals that are named here should be inserted in this section. Additionally, this extensive section should be classified according to the type of pollutant, e.g., 3.1.1 Dyes 3.1.2 Alcohols, phenols 3.1.3 Pesticides, etc etc. The authors should make an important effort to show in an instructive way this section. It is essential for the paper to be accepted.

Line 304: Please elaborate on the term water fertility, it is a good or bad consequence??? What does it generate?

Line 302-404 The same as in section 3.1. A comprehensive table including type of cyanobacteria, type of chemical, degradation rate, and reference for all the chemicals that are named here should be inserted in this section. Additionally, this extensive section should be classified according to the type of pollutant, e.g., 3.2.1 Dyes etc, etc. The authors should make an important effort to show in an instructive way this section. It is essential for the paper to be accepted.

Line 277-404: a couple of graphics should be added indicating on the one hand algae, and on the other hand cyanobacteria’s structure and mechanism of contaminant removal related to that microstructure

Line 405. A thorough review of techno-economic analyses (Section 4.1) and LCA (Section 4.2) approaches for these types of systems should be added. Including a comparative table with some other not so “sustainable” methods. For example, comparing this to activated sludge or other types of wastewater treatment from a techno-economic and lifecycle point of view.

Line 438: world's most serious (environmental???) challenges

Line 439: “microalgae and cyanobacteria is a sustainable “ Are the authors showing the LCA to ascertain this???

Line 449: Moreover, the relocation of novel pollutant-degrading bacterial genes into algae will 448 be applied to accelerate the degradation rate, breaking down a variety of organic contam- 449 inants. (insert reference here)

Line 456-457: Microalgae and cyanobacteria bioremedia- 456 tion of numerous organic contaminants is a sustainable and ecologically acceptable green 457 solution for treating polluted water. (This is the same sentence as line 439). Please check and rewrite all the conclusions

Line 460: Genetic engineering can boost algal 459 biodegradation capability and tolerance to various organic pollutants. (insert reference here) Is this the same as line 449??

Line 450-452: Enhancing the speed of phycoremediation of organic contaminants and shortening 450 the time required for decontamination will also require altering the physicochemical pa- 451 rameters of aquatic systems, such as temperature, pH, and nutrient availability.

Line 465- 468: To accelerate or- 465 ganic contaminant phycoremediation and reduce decontamination time, aquatic systems' 466 physicochemical characteristics such as temperature, pH, and nutrient availability must 467 be altered.

Aren’t this previous two sentences the same??

I would say that if the authors perform an extensive revision of the paper, it can be accepted.

Best regards

Author Response

Reviewer 3

Comments and Suggestions for Authors

The authors present a review of the use and applications of cyanobacteria and microalgae in wastewater treatment.

I would go directly so that the authors can understand why I would recommend a thorough revision and rewriting of most of the paper to be accepted:

Line 70: Please elaborate more on this sentence. It is not easy to understand as it is: “The adsorption method for the decolonization process was easy operation 70 and low cost [15]. »

While the adsorption technique is the most widely utilized method for decolonization due to its low costs and convenience of use [15].

Line 82 : « The toxicity of organometallic compounds is little understood, de- 82 spite being the most deadly substances on the market. » Reference here, please.

  • Done [2, 17]

Line 92-95 : were being the most widely used pesticides and herbicides in agriculture (rewrite please to give clear sense)

  •  

Line 97 : use synonym for bad : Having them in water is bad for ecosystems (harmful ??? toxic ???)

  •  

Line 118-123 : rewrite this sentence : These findings will help us better understand how phytoremediation can be achieved 118 using cyanobacteria. Using Cyanobacteria as a bioremediation tool in recycling. Also, with 119 the interest of elucidating the nature of this particular algal assemblage. To improve agri- 120 cultural processes, cyanobacteria can be used because of their high activity in photosyn- 121 thesis and the ability to separate the bioactive material from these algal groups Microalgae 122 species for phytoremediation of organic contaminants in aquatic settings are discussed by 123 many authors [26,27].

  • Done

Line 169-175: References 39,41 and 42 doesn’t seem to outline specifically the subject of biosurfactants from cyanobacteria and algae. I would recommend other references related to the production of such biomolecules specifficaly and making a mention of the types of molecules (biosurfactants that are produced). Besides, biosurfactant-producing algae may significantly enhance the biodegradation of hydrocarbon-polluted positions [39]. These biosurfactants are amphiphiles, thereby increasing the solubility of hydrophobic pollutants in water to accelerate hydrocarbon bioremediation [41]. PCB biodegradation by algae is dependent on their physicochemical and physiological the concentration and hydrophobicity of PCBs, the algae's biomass and physiological capacities, as well as the algae's membrane permeability [42].

References 41 &42 are modified.

 Baghour (2019) [39] reported that biosurfactant-producing microalgae may play an important role in the accelerated bioremediation of hydrocarbon-contaminated sites.

Line 175-176: Where is the 3 Title???

  1. Algae as organic biodegradator line 100

Line 185: that can (be used to) manufacture biodiesel [46,47].

  • Done

Line 210-211: Give sense to this sentence please: “Nitzschia sp. bioremediated phenanthrene (PHE) and fluoranthene (FLA) more rapidly 210 than Skeletonema costatum [61].”

  • The bioremediation rate of phenanthrene (PHE) and fluoranthene (FLA) by Nitzschia was faster than Skeletonema costatum [61].

Line 217: involved ?? erase in

  • Done

Line 228-235: The green microalgae Chlorella sp. 228 and Sc. obliquus exhibited biodegradation ability associated with many phenolic com- 229 pounds, which are listed by the U.S. Environmental Protection Agency “EPA” as priority 230 pollutants [71]. Pseudochlorococcum sp., Chlorella sp., and Chlamydomonas sp. are effectively 231 utilized to treat phenol-contaminated wastewater [69].

And repeat?????

The green microalgae Chlorella sp. 232 and S. obliquus showed biodegradation capacity in the presence of several phenolic com- 233 pounds that are identified as priority pollutants by the United States Environmental Protection Agency "EPA" [71].

  • Done
  • Line 277-300: A comprehensive table including type of algae, type of chemical, degradation rate, and reference for all the chemicals that are named here should be inserted in this section. Additionally, this extensive section should be classified according to the type of pollutant, e.g., 3.1.1 Dyes 3.1.2 Alcohols, phenols 3.1.3 Pesticides, etc etc. The authors should make an important effort to show in an instructive way this section. It is essential for the paper to be accepted.
  • Table 1& 2 (P5 line 206 & P9 390) containing the removal efficiency of organic pollutants using microalgae in laboratory-scale.
  • Line 304: Please elaborate on the term water fertility, it is a good or bad consequence??? What does it generate?
  • Fertility means an increase in the available nutrients (especially nitrogen) in water system which is called eutrophication. Eutrophication causes algae to grow faster than ecosystems can handle which can severely reduce or eliminate oxygen in the water, leading to illnesses in fish and the death of large numbers of fish.

Line 302-404 The same as in section 3.1. A comprehensive table including type of cyanobacteria, type of chemical, degradation rate, and reference for all the chemicals that are named here should be inserted in this section. Additionally, this extensive section should be classified according to the type of pollutant, e.g., 3.2.1 Dyes etc, etc. The authors should make an important effort to show in an instructive way this section. It is essential for the paper to be accepted.

  • Table 2 (P9 390).

Line 277-404: a couple of graphics should be added indicating on the one hand algae, and on the other hand cyanobacteria’s structure and mechanism of contaminant removal related to that microstructure.

Done as illustrated in Figure 1.

Line 405. A thorough review of techno-economic analyses (Section 4.1) and LCA (Section 4.2) approaches for these types of systems should be added. Including a comparative table with some other not so “sustainable” methods. For example, comparing this to activated sludge or other types of wastewater treatment from a techno-economic and lifecycle point of view.

- Table 3 P10 L434

Line 438: world's most serious (environmental???) challenges

-Done.

Line 439: “microalgae and cyanobacteria is a sustainable “Are the authors showing the LCA to ascertain this???

  • There are many researches for us concerning study the life cycle of different microalgal species like
  • Mostafa M EL-Sheekh, Mohamed Y Bedaiwy, Mohamed E Osman and Mona M Ismail Influence of Molasses on Growth, Biochemical Composition and Ethanol Production of the Green Algae Chlorella Vulgaris and Scenedesmus Obliquus. Journal of Agricultural Engineering and Biotechnology. 2014. 2(2): 20-28.
  • Mostafa M EL-Sheekh, Mohamed Y Bedaiwy, Mohamed E Osman and Mona M Ismail Mixotrophic and heterotrophic growth of some microalgae using extract of fungal-treated wheat bran. International Journal Of Recycling of Organic Waste in Agriculture 2012, 1:12
  • Algae and /or bacteria are sustainable resources for many product innovation; they are primary producers who can fulfill abovementioned needs due to their ability to efficiently harvest solar energy and convert it into biomass by simple utilization of CO2, water and nutrients (Pathak et al., 2018). Moreover they exist in various aquatic and terrestrial ecosystems, and require no arable land for production (Hopes and Mock 2015).
  • Pathak  , Rajneesh,  Maurya PK., Singh S.P,  Häder D-P and Sinha RP(2018). Cyanobacterial Farming for Environment Friendly Sustainable Agriculture Practices: Innovations and Perspectives. Front. Environ. Sci., 28 February 2018. https://doi.org/10.3389/fenvs.2018.00007.

-       Hopes A, Mock T (2015) Evolution of microalgae and their adaptations in different marine ecosystems. Wiley, New York. https://doi.org/10.1002/9780470015902.a0023744

Line 449: Moreover, the relocation of novel pollutant-degrading bacterial genes into algae will 448 be applied to accelerate the degradation rate, breaking down a variety of organic contam- 449 inants. (insert reference here).

- This is suggestion for future studies from the author's point of view.

- Line 456-457: Microalgae and cyanobacteria bioremedia- 456 tion of numerous organic contaminants is a sustainable and ecologically acceptable green 457 solution for treating polluted water. (This is the same sentence as line 439). Please check and rewrite all the conclusions

- It is modified.

Line 460: Genetic engineering can boost algal 459 biodegradation capability and tolerance to various organic pollutants. (insert reference here) Is this the same as line 449??

  • Conclusion is rewritten and modified. Usually conclusion does not contain references.
  • Genetic engineering paragraph with references is illustrated in lines 390-404.

  • Line 450-452: Enhancing the speed of phycoremediation of organic contaminants and shortening 450 the time required for decontamination will also require altering the physicochemical pa- 451 rameters of aquatic systems, such as temperature, pH, and nutrient availability.
  • Conclusion is modified.

  • Line 465- 468: To accelerate or- 465 ganic contaminant phycoremediation and reduce decontamination time, aquatic systems' 466 physicochemical characteristics such as temperature, pH, and nutrient availability must 467 be altered.Aren’t this previous two sentences the same??
  • Done

I would say that if the authors perform an extensive revision of the paper, it can be accepted.

Round 2

Reviewer 1 Report

Dear Sir,

I think the authors have made significant improvements, addressing most of my previous comments. Therefore, I think the manuscript is now fit for publication in this journal.

Kind regards.

Author Response

Thank You

Reviewer 2 Report

Acceptance

Author Response

Thank You

Reviewer 3 Report

Dear Editors,

The authors performed most of the suggested changes and the paper is improved. Although, still some changes that are indicated as "done", were not performed.

I would suggest the authors to perform these changes so that the paper could be approved

Best regards

Line 82 : « The toxicity of organometallic compounds is little understood, de- 82 spite being the most deadly substances on the market. » Reference here, please.

  • Done [2, 17] Was not

Line 118-123 : rewrite this sentence : These findings will help us better understand how phytoremediation can be achieved 118 using cyanobacteria. Using Cyanobacteria as a bioremediation tool in recycling. Also, with 119 the interest of elucidating the nature of this particular algal assemblage. To improve agri- 120 cultural processes, cyanobacteria can be used because of their high activity in photosyn- 121 thesis and the ability to separate the bioactive material from these algal groups Microalgae 122 species for phytoremediation of organic contaminants in aquatic settings are discussed by 123 many authors [26,27].

  • Done Was not

 Line 82 : « The toxicity of organometallic compounds is little understood, de- 82 spite being the most deadly substances on the market. » Reference here, please.

  • Done [2, 17] Was not

Line 185: that can (be used to) manufacture biodiesel [46,47].

  • DoneWas not

Line 331-335 It is repeated.

Both C. vulgaris and Sc. obliquus have been discovered as potential 331 biosystems for crude oil degradation due to their rapid growth rates and capacity to bio- 332 degrade oil under heterotrophic conditions using waste oil as the sole carbon source [82- 333 84].

Both C. vulgaris and Sc. obliquus have been discovered as potential biosystems for 334 crude oil degradation due to their rapid growth rates and capacity to biodegrade oil under 335 heterotrophic conditions using waste oil as the sole carbon source [82-84]

Line 439: “microalgae and cyanobacteria is a sustainable “Are the authors showing the LCA to ascertain this???

  • There are many researches for us concerning study the life cycle of different microalgal species like
  • Add these references to the text:
  • Mostafa M EL-Sheekh, Mohamed Y Bedaiwy, Mohamed E Osman and Mona M Ismail Influence of Molasses on Growth, Biochemical Composition and Ethanol Production of the Green Algae Chlorella Vulgaris and Scenedesmus Obliquus. Journal of Agricultural Engineering and Biotechnology. 2014. 2(2): 20-28.
  • Mostafa M EL-Sheekh, Mohamed Y Bedaiwy, Mohamed E Osman and Mona M Ismail Mixotrophic and heterotrophic growth of some microalgae using extract of fungal-treated wheat bran. International Journal Of Recycling of Organic Waste in Agriculture 2012, 1:12
  • Algae and /or bacteria are sustainable resources for many product innovation; they are primary producers who can fulfill abovementioned needs due to their ability to efficiently harvest solar energy and convert it into biomass by simple utilization of CO2, water and nutrients (Pathak et al., 2018). Moreover they exist in various aquatic and terrestrial ecosystems, and require no arable land for production (Hopes and Mock 2015).
  • Pathak  , Rajneesh,  Maurya PK., Singh S.P,  Häder D-P and Sinha RP(2018). Cyanobacterial Farming for Environment Friendly Sustainable Agriculture Practices: Innovations and Perspectives. Front. Environ. Sci., 28 February 2018. https://doi.org/10.3389/fenvs.2018.00007.

-       Hopes A, Mock T (2015) Evolution of microalgae and their adaptations in different marine ecosystems. Wiley, New York. https://doi.org/10.1002/9780470015902.a0023744

Author Response

The authors performed most of the suggested changes and the paper is improved. Although, still some changes that are indicated as "done", were not performed.

I would suggest the authors to perform these changes so that the paper could be approved

Best regards

Line 82 : « The toxicity of organometallic compounds is little understood, de- 82 spite being the most deadly substances on the market. » Reference here, please.

  • Done [2, 17] Was not

Egorova K.S. and Ananikov VP. (2017) Toxicity of Metal Compounds: Knowledge and Myths. Organometallics 2017, 36, 4071−4090.

House, J. E. and K. A. House (2016). Chapter 22 - Organometallic Compounds. Descriptive Inorganic Chemistry (Third Edition). J. E. House and K. A. House. Boston, Academic Press: 371-393.

Line 118-123 : rewrite this sentence : These findings will help us better understand how phytoremediation can be achieved 118 using cyanobacteria. Using Cyanobacteria as a bioremediation tool in recycling. Also, with 119 the interest of elucidating the nature of this particular algal assemblage. To improve agri- 120 cultural processes, cyanobacteria can be used because of their high activity in photosyn- 121 thesis and the ability to separate the bioactive material from these algal groups Microalgae 122 species for phytoremediation of organic contaminants in aquatic settings are discussed by 123 many authors [26,27].

  • Done 

Line 185: that can (be used to) manufacture biodiesel [46,47].

  • Done

Line 331-335 It is repeated.

Both C. vulgaris and Sc. obliquus have been discovered as potential 331 biosystems for crude oil degradation due to their rapid growth rates and capacity to bio- 332 degrade oil under heterotrophic conditions using waste oil as the sole carbon source [82- 333 84].

Both C. vulgaris and Sc. obliquus have been discovered as potential biosystems for 334 crude oil degradation due to their rapid growth rates and capacity to biodegrade oil under 335 heterotrophic conditions using waste oil as the sole carbon source [82-84]

  • Removed

Line 439: “microalgae and cyanobacteria is a sustainable “Are the authors showing the LCA to ascertain this???

  • There are many researches for us concerning study the life cycle of different microalgal species like
  • Add these references to the text:
  • Mostafa M EL-Sheekh, Mohamed Y Bedaiwy, Mohamed E Osman and Mona M Ismail Influence of Molasses on Growth, Biochemical Composition and Ethanol Production of the Green Algae Chlorella Vulgaris and Scenedesmus Obliquus. Journal of Agricultural Engineering and Biotechnology. 2014. 2(2): 20-28. Line 173, 279
  • Mostafa M EL-Sheekh, Mohamed Y Bedaiwy, Mohamed E Osman and Mona M Ismail Mixotrophic and heterotrophic growth of some microalgae using extract of fungal-treated wheat bran. International Journal Of Recycling of Organic Waste in Agriculture 2012, 1:12. P. 4; L. 170
  • Algae and /or bacteria are sustainable resources for many product innovation; they are primary producers who can fulfill abovementioned needs due to their ability to efficiently harvest solar energy and convert it into biomass by simple utilization of CO2, water and nutrients (Pathak et al., 2018). P.4; L. 176.
  • they exist in various aquatic and terrestrial ecosystems, and require no arable land for production (Hopes and Mock 2015). P.6; L.284
  • Pathak  , Rajneesh,  Maurya PK., Singh S.P,  Häder D-P and Sinha RP(2018). Cyanobacterial Farming for Environment Friendly Sustainable Agriculture Practices: Innovations and Perspectives. Front. Environ. Sci., 28 February 2018. https://doi.org/10.3389/fenvs.2018.00007.

-       Hopes A, Mock T (2015) Evolution of microalgae and their adaptations in different marine ecosystems. Wiley, New York. https://doi.org/10.1002/9780470015902.a0023744
